# Genome-Wide Identification of Catalase Gene Family and the Function of *SmCAT4* in Eggplant Response to Salt Stress

**DOI:** 10.3390/ijms242316979

**Published:** 2023-11-30

**Authors:** Lei Shen, Xin Xia, Longhao Zhang, Shixin Yang, Xu Yang

**Affiliations:** College of Horticulture and Landscape Architecture, Yangzhou University, Yangzhou 225009, China; shenlei07@yzu.edu.cn (L.S.); mz120221425@stu.yzu.edu.cn (X.X.); mz120231508@stu.yzu.edu.cn (L.Z.); mz120221438@stu.yzu.edu.cn (S.Y.)

**Keywords:** eggplant (*Solanum melongena*), catalase, *SmCAT4*, salt stress

## Abstract

Salinity is an important abiotic stress, damaging plant tissues by causing a burst of reactive oxygen species (ROS). Catalase (CAT) enzyme coded by *Catalase (CAT)* genes are potent in reducing harmful ROS and hydrogen peroxide (H_2_O_2_) produced. Herein, we performed bioinformatics and functional characterization of four *SmCAT* genes, retrieved from the eggplant genome database. Evolutionary analysis *CAT* genes revealed that they are divided into subgroups I and II. The RT-qPCR analysis of *SmCAT* displayed a differential expression pattern in response to abiotic stresses. All the CAT proteins of eggplant were localized in the peroxisome, except for SmCAT4, which localized in the cytomembrane and nucleus. Silencing of *SmCAT4* compromised the tolerance of eggplant to salt stress. Suppressed expression levels of salt stress defense related genes *SmTAS14* and *SmDHN1*, as well as increase of H_2_O_2_ content and decrease of CAT enzyme activity was observed in the *SmCAT4* silenced eggplants. Our data provided insightful knowledge of *CAT* gene family in eggplant. Positive regulation of eggplant response to salinity by *SmCAT4* provides resource for future breeding programs.

## 1. Introduction

Due to their sessile nature, plants must endure various kinds of stresses such as heat [1,2], cold [3,4,5], salt [6,7], drought [8], heavy metal [9], insect infestation [10,11], and pathogens [12,13]. Reactive oxygen species (ROS) including hydrogen peroxide (H_2_O_2_), oxygen free radical or superoxide (O^2−^), hydrogen free radical (·OH), and non-radical singlet oxygen (^1^O_2_) act as signaling molecules that modulate plant homeostatic mechanism [14,15]. However, excessive accumulation of ROS can damage cells, eventually affect growth, development, or, in severe cases, plant death. Plant have evolved sophisticated mechanisms such as enzymatic or non-enzymatic detoxification systems to alleviate the damage of ROS. The antioxidant enzyme machinery including peroxidase (POD), catalase (CAT), superoxide dismutase, ascorbate peroxidase (APX), glutathione S transferase (GST), glutathione peroxide (GPX), dehydroascorbate reductase (DHAR), and glutathione reductase (GR) [16,17,18,19]. The POD, CAT, APX, and SOD act as a protector by scavenging H_2_O_2_, and enhance plant immunity against abiotic and biotic stresses [16].

CAT converging with other components of antioxidant machinery is key in plant stress biology [20]. CAT can effectively scavenge H_2_O_2_, further relieving the damaging effects of oxidative stress. CAT is a tetrameric heme protein composed of four subunits. They are mainly distributed in the peroxisome, glyoxylate circulator, and cytoplasm Apart from that, a small amount of distribution in mitochondria and chloroplast was also recorded [21]. CATs are encoded by multiple genes with numbers varying across different species. For instance, 3 CATs in *Arabidopsis thaliana* [22], 3 CATs in rice (*Oryza sativa*) [23], 7 CATs in cotton (*Gossypium hirsutum*) [24], 10 CATs in wheat (*Triticum aestivum*) [20], and 14 CATs in *Brassica napus* [25] were mapped. CATs regulated all aspects of plant growth and development by maintaining intracellular redox homeostasis [26]. ABA INSENSITIVE 5 (ABI5) activated *AtCAT1* expression by the directly binding to the promoter of *AtCAT1* to affect the ROS homeostasis, and then promoted the seed germination in Arabidopsis [27]. The hot pepper (*Capsicum annuum*) *CaCAT1* and *CaCAT2* were involved in the modulation of circadian rhythms and their expression was tightly organ-specific [28]. In maize, the P31 protein of chlorotic mottle virus (MCMV) interacts with ZmCAT1 and the P31-ZmCAT1 cascade further suppresses the expression of *PR1* gene thus enhancing the accumulation of viral load. Similarly, the interaction of *Phytophthora sojae* effector PsAvh113 with transcription factor GmDPB decreases the expression level of *GmCAT1*, thereby decreasing plant resistance to *Phytophthora* [29]. In rice, E3 ligase APIP6 degraded catalase OsCATC to negatively regulate rice innate immunity against blast fungus *Magnaporthe oryzae* [30]. These findings indicate that CATs play important roles in plants against various pathogenic microorganism. CAT facilitation of plant immunity against abiotic stresses are well studied. Induced expression of *MeCu*/*ZnSOD* and *MeCAT1* enhanced the tolerance of cassava against cold and drought stresses [31]. WD40 protein TaWD40-4b.1C interacts with TaCAT1A and TaCAT3A, promote their oligomerization and enzyme activities. This interaction amplifies the wheat response to drought stress by reducing H_2_O_2_ level [32]. Although many studies reported that CATs play an important role in plant response to abiotic stresses, the functions and regulatory mechanism of CATs remain unclear.

Eggplant (*Solanum melongena*) is a popular *Solanaceae* vegetable cultivated all around the world [33]. The growth, development, and yield of eggplant were vulnerable to various environmental stresses such as salt, drought, high temperature, and cold stress. The continuous environmental shift encourages the breeding of new eggplant varieties with high tolerance to multiple abiotic stresses. In the present study, a comprehensive genome-wide analysis of eggplant *CAT* gene family was performed. A total of four *CAT* genes were isolated from the eggplant genome database. A detailed transcript expression analysis of *CATs* in eggplant was carried out. Virus induced gene silencing (VIGS) assay showed that *SmCAT4* plays a positive role in eggplant response to salt stress.

## 2. Results

### 2.1. Identification and Physicochemical Property Analysis of Catalase Proteins in Eggplant

According to the amino acid sequences of Arabidopsis *CAT* gene family members, we identified four *CAT* genes (*SmCAT1*, *SmCAT2*, *SmCAT3*, and *SmCAT4*) in the eggplant genome by using TBtools software (version 2.019) and the NCBI website. We further analyzed the physicochemical properties of SmCATs (Table 1). *SmCAT1*-*3* encoded 492 amino acids with 1479 bp, while *SmCAT4* encoded 491 amino acids with 1476 bp. The molecular weight, instability index, aliphatic index, and average hydropathicity of SmCAT1-4 are 56,545~56,996 Da, 38.18~40.25, 68.94~71.91, and −0.582~−0.515, respectively. We also found that the theoretical pI and grand average of hydropathicity of SmCAT1-4 proteins are 6.80~7.31 and −0.582~−0.515, respectively, implying that SmCAT1-4 proteins are hydrophilic non-transmembrane proteins. In addition, the prediction results of subcellular localization showed that SmCAT1-4 proteins are located in the peroxisome.

### 2.2. Phylogenic Analysis of Catalase Proteins

We constructed the phylogenetic tree by using MEGA 7.0 software (version 7.0.26) to investigate the phylogenetic relationships of the *CAT* gene family members in eggplant with their homologs from Arabidopsis, tomato (*Solanum lycopersicum*), and soybean (*Glycine max*). All *CAT* genes in the phylogenetic tree are divided into subgroups Ⅰ and Ⅱ (Figure 1). The subgroup Ⅰ consists of four *CATs* (*OsCATA-D*) of rice, *AtCAT3*, and three *SmCATs* (*SmCAT1*, *SmCAT2*, *SmCAT4*). The subgroup Ⅱ consists of four *CATs* (*GmCAT1/2-5*) of soybean, *SmCAT3*, and *AtCAT1* and *AtCAT2*. In subgroup Ⅰ, four soybean *CATs* were grouped into the same subgroup as the two members *AtCAT1* and *AtCAT2* of Arabidopsis, implying that their protein sequences have high similarity. Similarly, four *OsCATs* were divided into subgroup Ⅱ with three *SmCATs*, suggesting that the protein sequences of *OsCATs* have high similarity with the *SmCATs*.

### 2.3. Sequence Analysis of CAT Genes in Eggplant

The visualization of the chromosome location of eggplant *CATs* was performed using TBtools software (version 2.019) [34]. The *SmCAT* genes are scattered over chromosomes 02, 04, and 05. We also found that *SmCAT1* and *SmCAT4* were located very close to each other on chromosome 5 (Figure 2a). We next investigated the gene structures of *CATs* in eggplant by identifying their exon-intron orientation. *SmCAT1-3* has eight exons, one unique 5′-UTR and 3′-UTR regions, while *SmCAT4* has seven exons but no 5′-UTR and 3′-UTR regions. The size of *SmCTAs* is about 4~5 kb except for *SmCAT4*, which is close to 2 kb (Figure 2b). The conserved domain of SmCATs was predicted by the SMART website and visualized by DOG 2.0 software. All SmCAT proteins have one Catalase and Catalase-rel (catalase-related immune response) domain, and the locations of these two domains within four SmCAT proteins are very close to each other (Figure 2c). The motifs in SmCAT proteins were analyzed by searching the MEME website. The result showed that all SmCAT proteins contain 10 identical motifs. The majority of them are at least 50 amino acids in length, except for motifs 9 and 10, which have lower sizes (Figure 2d). We further predicted the *cis*-elements within the promoters of *SmCATs* by searching the PlantCARE website and visualizing by TBtools software. The observed *cis*-elements include light responsiveness or unknown functions. Multiple phytohormone responsive *cis*-elements, such as ethylene (ET) response element (ERE), salicylic acid (SA) response element (TCA-element), jasmonic acid methyl ester (MeJA) response element (TGACG-motif), and ABA response element (ABRE) were identified. Important transcription factor binding sites such as WRKY transcription factor binding site (W-box), MYB binding site (MYB), MYC binding site (MYC), and bZIP binding site (A-box and G-box) as well as stress responsive *cis*-elements including low-temperature response element (LTR) and anaerobic induction element (ARE) were harbored in the promoter of these four *SmCATs* (Figure 2e). In addition, the tertiary structures of four SmCAT proteins were analyzed by searching the SWISS-MODEL website. We found similarities between SmCAT1, SmCAT3, SmCAT4, and A0A5J5B3V7.1.A catalase model of 80.08, 90.85, and 77%, respectively. The similarity between SmCAT2 and B9S6U0.1.A catalase model is 90.2% (Figure 2f).

### 2.4. Collinearity Analysis of Eggplant CAT Genes and Their Homologs from Arabidopsis and Tomato

Tandem duplications and segmental duplications significantly contributed to the expansion of gene families [35]. So, we analyzed the gene collinearity relationship of *SmCAT* genes and their homologs from Arabidopsis and tomato by using TBtools software. Two duplicated segments (*SmCAT4* and *SmCAT3*, *SmCAT3*, and *SmCAT2*) were identified in the eggplant genome (Figure 3a). Additionally, we found that tandem duplication events of *CAT* gene family members did not occur in the eggplant genome. To further explore the evolutionary relationship of the *CAT* genes between eggplant, Arabidopsis, and tomato, we investigated the collinearity of *SmCATs* with that of Arabidopsis and tomato. We found that *SmCAT2* showed a synteny relationship with *AtCAT1* of Arabidopsis and *SlCAT2* and *SlCAT3* of tomato, respectively. Meanwhile, both *SmCAT3* and *SmCAT4* exhibited a synteny relationship with *AtCAT1* of Arabidopsis and *SlCAT1*, *SlCAT2*, and *SlCAT3* of tomato, respectively (Figure 3b).

### 2.5. Expression Analysis of Eggplant CATs under Abiotic Stress Treatment and in Different Tissues

We analyzed the expression of *SmCATs* under various abiotic stresses, including dehydration, salt, high temperature (HT), low temperature (LT), H_2_O_2_, and ABA by real-time quantitative PCR (RT-qPCR) assay (Figure 4). In response to dehydration stress, the expression levels of four eggplant *CATs* were significantly up-regulated (Figure 4a). Under salt stress, the expression levels of *SmCAT3* and *SmCAT4* were significantly up-regulated, whereas *SmCAT1* expression level was significantly down-regulated. On the other hand, the expression levels of *SmCAT2* displayed no obvious change (Figure 4b). To HT treatment, the expression levels of *SmCAT1* and *SmCAT2* were significantly down-regulated while *SmCAT4* were significantly up-regulated (Figure 4c). The expression levels of *SmCAT1*, *SmCAT2*, and *SmCAT4* showed a trend of significant up-regulation and then down, while the *SmCAT3* were significantly down-regulated in response to LT treatment (Figure 4d). Under H_2_O_2_ treatment, *SmCAT1*, *SmCAT2*, and *SmCAT3* showed a significant up-regulation expression trajectory, but *SmCAT4* were down-regulated (Figure 4e). ABA treatment could induce the expression levels of *SmCAT1*, *SmCAT2*, and *SmCAT3*—but not *SmCAT4* (Figure 4f).

We further investigated the expression levels of eggplant *CATs* in the different tissues of seedlings and mature plants (Figure 5). The expression levels of *SmCAT1*, *SmCAT2*, *SmCAT3*, and *SmCAT4* were highest in the young leaf (YL), root (RT), and stem (ST) of seedlings. All four *SmCATs* showed high expressions in the sepal (SE), fully expanded leaf (FEL), flower (FL), and ST of mature eggplant.

### 2.6. Subcellular Localization of Eggplant CAT Proteins

The subcellular location was predicted using an online server and it revealed that all SmCATs are resided in the peroxisomes (Table 1). To test this result, we performed the transient expression of *SmCATs* in the leaves of *Nicotiana benthamiana* via Agrobacterium-mediated transformation. The full-length ORF of eggplant *SmCATs* were cloned into plant overexpression vector pBinGFP2 (Figure 6a). After 48 h infiltration, we observed the fluorescence signal by using a laser scanning confocal microscope. We found that the green fluorescence signal of all eggplant CAT proteins appears in the peroxisome, except for SmCAT4-GFP whose green fluorescence signal appeared in the cytomembrane and nucleus, while the green fluorescence signal of GFP exists in the whole cells (Figure 6b).

### 2.7. Recombinant SmCAT4 Enzyme Activity Analysis

We found that the expression of *SmCAT4* was up-regulated by dehydration, salt, HT, and LT treatment (Figure 4), indicating that *SmCAT4* may play an essential role in eggplant response to the above abiotic stresses. Hence, we selected *SmCAT4* to explore its functions in eggplant response to salt stress. We first analyzed *SmCAT4* enzyme activity by recombinant CAT enzyme assay in vitro. The recombinant SmCAT4-GST proteins or GST proteins were expressed by prokaryotic expression assay and purified by GST magnetic beads and then verified by SDS-PAGE assay (Figure 7a). To detect the CAT enzyme activity of SmCAT4, we tested the capacity of recombinant SmCAT4-GST proteins to increase H_2_O_2_ scavenging. Compared to the control (GST proteins), SmCAT4-GST exhibited distinctly H_2_O_2_ scavenging capacity (Figure 7b). These data indicate that SmCAT4 exhibits CAT enzyme activity to scavenge H_2_O_2_.

### 2.8. Silencing of SmCAT4 Enhanced Susceptibility of Eggplant to Salt Stress

To investigate the function of *SmCAT4* in eggplant, we assessed the effect of *SmCAT4* silencing on the tolerance of eggplant to salt stress using VIGS assay. We first detected the silencing efficiency of *SmCAT4* by RT-qPCR assay. Compared to the control plants (TRV:*00*), the expression level of *SmCAT4* in the roots of *SmCAT4*-silenced (TRV:*SmCAT4*) plants was significantly reduced by approximately 70% under salt stress treatment (Figure 8a). Silencing of *SmCAT4* enhanced the susceptibility of eggplant to salt stress and exhibiting a lower survival rate compared to the control plants at 48 h post salt stress treatment (Figure 8b,c). In addition, we also found that silencing of *SmCAT4* significantly down-regulated the expression levels of salt stress defense-related marker genes *SmATS14* and *SmDHN1* (Figure 8d), accompanied by an increase of H_2_O_2_ content and a decrease of CAT enzyme activity (Figure 8e). These data imply that *SmCAT4* positively functions in eggplant response to salt stress.

## 3. Discussion

Excessive accumulation of ROS caused serious damage to the stability of the plasma membrane in plant cells. The ROS scavenging system, comprising enzymatic and non-enzymatic components [36], is key to plant survival under adverse conditions. CATs, one of the most important ROS scavenging enzymes, has been involved in regulating plant response to abiotic stresses such as salt [37,38], drought [31,39,40,41], cold [31], heat [42,43], osmotic stress [37], and biotic stresses including pathogenic microorganisms attack [30,44,45,46,47,48,49,50], and insect invasion [51]. In addition, growth and development [27,52] and senescence [53] are also regulated by CATs in conjunction with growth-related hormones. However, the underlying mechanisms of CATs in plants involved in the regulation of salt stress resistance largely remain unclear. In this study, we identified four *CAT* genes in the eggplant genome, analyzed their sequences and structures and explored their expressions under NaCl, dehydration, LT, HT, H_2_O_2_, and ABA treatment. We further provided evidence that *SmCAT4* positively functions in eggplant response to salt stress.

ROS has dual functions, and the causal link between ROS production and stress tolerance is not as straightforward as one may expect [54]. Upon normal circumstances, ROS acts as a second messenger to integrate signaling pathways involved in the regulation of growth, development, gravitropism, phytohormone signal transduction, defense response, and many other physiological processes [55,56,57,58,59]. When ROS production is excessive, uncontrolled oxidation ultimately leads to cellular damage and even cell death. To avoid excessive cell damage, excessive accumulation of ROS should be scavenged, and the antioxidant defenses must keep ROS under control. Therefore, analyzing ROS scavenging is beneficial for a better understanding of the functions and mechanisms of ROS in various physiological processes in plants. CATs act as the essential ROS scavenging enzyme and play an important role in plant response to salt stress [37,38]. However, to date, the information and functions of *CAT* genes in eggplant remain indistinct. Herein, we identified four *CAT* genes in the eggplant genome, which exhibited similar physicochemical properties (Table 1). By generating a phylogenetic tree, the *CAT* genes of eggplant mainly clustered with four rice *CAT* genes and *AtCAT3* apart from *SmCAT3*, while *SmCAT3* exhibited the closest phylogenetic relationship with *AtCAT1*, *AtCAT2*, and four soybean *CAT* genes (Figure 1), implying that there is likely a functional differentiation between *SmCAT3* and the three other *CAT* genes in eggplant. We analyzed sequences and structures of *CAT* genes in eggplant and found that four eggplant *CAT* genes were located in chromosomes 02, 04, and 05 (Figure 2a). Notably, *SmCAT1* and *SmCAT4* were located very close to each other in chromosome 05, which was similar to *AtCAT1* and *AtCAT3* [14], suggesting that *SmCAT1* and *SmCAT4* most likely evolved from a common ancestral gene and had similar functions. We analyzed the gene structures of eggplant *CAT* genes and found that *SmCAT1-3* has eight exons, while *SmCAT4* has seven exons (Figure 2b). Each of the four eggplant CAT proteins has conserved Catalase and Catalase-rel domain as well as 10 identical conservative motifs (Figure 2c,d) but did not have a PP-Binding domain such as Arabidopsis CAT proteins [14]. Apart from the *cis*-elements related to light response, we counted the *cis*-elements related to phytohormone response, transcription factors binding, and stress response within the promoters of four eggplant *CAT* genes by searching the PlantCARE website and found the phytohormone response-related *cis*-elements including SA response-related element CAT-element, ET response-related element ERE, ABA response-related element ABRE, and MeJA response-related element TGACG-motif, and transcription factors binding elements such as HD-Zip Ⅰ, W-box, MYB, MYC, G-box, and A-box, as well as stress response-related elements LTR, ARE, and STRE within the promoters of these four *CAT* genes (Figure 2e). Abundant *cis*-elements within the eggplant *CAT* gene promoters implied that *CAT* genes may be involved in multiple physiological processes. Gene duplication events act as a primary contributor to gene expansion, leading to the evolution and diversification of genes in plants [60]. In general, the functional relationships of homologs within the same species or in interspecific species reveal their conserved functions and the co-evolutionary origins of members within subgroups [61]. In eggplant, there are two tandem gene clusters, including *SmCAT3* and *SmCAT2* as well as *SmCAT3* and *SmCAT4*, but segmentally duplicated gene pairs do not exist (Figure 3a), indicating that *CAT* genes of eggplant have not been duplicated in large fragments during the evolutionary process, and this may be the reason that the eggplant *CAT* family has fewer members. The collinear analysis displayed that three *CAT* genes (*SmCAT2-4*) from eggplant were orthologous to Arabidopsis (*AtCAT1*) and tomato (*SlCAT2-4*), respectively (Figure 3b), suggesting that these genes may be functionally similar.

Accumulating pieces of evidence demonstrated that *CAT* genes have functions in plant response to abiotic stresses [31,38,39]. We detected the transcript expression levels of four eggplant *CAT* genes under stress, dehydration, LT, HT, H_2_O_2_, and ABA treatment and found that the transcript expression levels of the four *CAT* genes from eggplant were up- or down-regulated to different extents, and some *CAT* genes expression did not have obvious change (Figure 4). Notably, transcript expression levels of *SmCAT4* were significantly up-regulated by salt, dehydration, LT, HT, and H_2_O_2_ treatment, suggesting that *SmCAT4* may play an important role in eggplant against salt, dehydration, LT, and HT stresses. In addition, we tested the transcript expression levels of *SmCAT1-4* in the different tissues of seedlings or mature plants, and in seedlings, the expression of *SmCAT1* and *SmCAT3* was higher than that of in stem and young leaf (YL), while *SmCAT2* and *SmCAT4* were highly expressed in stem (ST) and root (RT), respectively. In mature plants, *SmCAT1-4*, respectively, were highly expressed in sepal (SE), fully expanded leaf (FEL), flower (FL), and stem (ST) (Figure 5). Previous studies showed that CAT proteins are mainly located in the peroxisome, glyoxylate circulator and cytoplasm [21,62]. Although we predicted the subcellular localization of these four CAT proteins, which localized in the peroxisome, we provided evidence that eggplant CAT1-3 proteins located in the peroxisome and SmCAT4 localized in the cytomembrane and nucleus (Figure 6). Similar studies revealed that *Pinellia ternata* catalase protein PtCAT2 localized in the cytoplasm and membrane [63]. Sugarcane catalase protein ScCAT2 is distributed in the nucleus, plasma membrane, and cytoplasm in the epidermal cells of *Nicotiana benthamiana* leaves [47]. These reports imply that not all CAT proteins function in peroxisomes. To explore the functions of *CAT* genes in eggplant’s response to abiotic stresses, we selected SmCAT4 to further analyze its function in eggplant’s response to salt stress. Silencing of *SmCAT4* decreased the tolerance of eggplant to salt stress (Figure 8b) and significantly down-regulated the transcript expression levels of salt stress defense-related genes *SmATS14* and *SmDNH1* (Figure 8d), accompanied by an increase of H_2_O_2_ content and a decrease of the enzyme activities of CAT (Figure 8e). In addition, we tested the CAT enzyme activity of SmCAT4 in vitro and found that SmCAT4 could scavenge H_2_O_2_ in vitro, indicating that SmCAT4 exhibits CAT enzyme activity. Likewise, the OsCAT3 protein has an obvious ability to remove H_2_O_2_ in vitro [23]. These data indicate that *SmCAT4* positively functions in eggplant response to salt stress via scavenging H_2_O_2_. Previous studies showed that Arabidopsis CAT2 promotes LAP2 hydrolysis activity with leucine-4-methylcoumaryl-7-amides as a substrate by interacting with LAP2 in vivo and in vitro to confer Arabidopsis salt and osmotic stress tolerance [37].

In conclusion, we identified four *CAT* genes in the eggplant genome, analyzed their sequences, structures and expressions, and provided evidence that SmCAT4 has an obvious ability to remove H_2_O_2_ in vitro and positively functions in eggplant response to salt stress.

## 4. Material and Methods

### 4.1. Identification of CAT Genes in Solanum melongena Genome

The amino acid sequences of three Arabidopsis CAT obtained from The Arabidopsis Information Resource database (https://www.arabidopsis.org/, accessed on 10 September 2023) were acted as reference sequences [22], and the members of CAT genes in eggplant genome were identified by combining TBtools software [34], National Center for Biotechnology Information (https://www.ncbi.nlm.nih.gov/, accessed on 10 September 2023) website, and Eggplant Genome Database (http://eggplant-hq.cn/Eggplant/home/index, accessed on 10 September 2023).

### 4.2. Physicochemical Properties Analysis of CAT Proteins in Eggplant

The physicochemical properties of eggplant CATs protein amino acid sequences, including total number of atoms, instability index, aliphatic index, average of hydropathicity, and theoretical pI, were analyzed by searching the ProtParam Expasy (https://web.expasy.org/protparam/, accessed on 11 September 2023) website [64]. The subcellular localization prediction of eggplant CATs was carried out by searching the Plant-mPLoc website (http://www.csbio.sjtu.edu.cn/bioinf/plant-multi/, accessed on 11 September 2023) with their amino acid sequences.

### 4.3. Multiple Sequence Alignment and Phylogenetic Analysis

The amino acid sequences of CAT members in soybean (*Glycine max*) and rice were respectively downloaded from SoyBase (https://soybase.org/, accessed on 10 September 2023) database [29] and The Rice Annotation Project (RAP) (https://rapdb.dna.affrc.go.jp/, accessed on 10 September 2023) database [14]. Multiple sequence alignment and phylogenetic analysis were carried out by using MEGA 7 (version. 7.0.26) software. The evolutionary tree was constructed by MEGA 7 using the neighbor-joining (NJ) method with 1000 bootstrap replications [65].

### 4.4. Analysis of Chromosomal Location, Gene Structure, Conserved Domain and Motif, cis-Elements, and Tertiary Structure of Eggplant CATs

The chromosomal locations and gene structures of eggplant CATs were visualized by using TBtools software with Gene Location Visualize from the GTF/GFF functional module and Visualize Gene Structure (Basic), respectively. The conserved domain of eggplant CATs was predicted by searching the SMART (http://smart.embl-heidelberg.de/, accessed on 11 September 2023) website and visualized by DOG 2.0 software [66]. The conserved motifs of eggplant CATs were predicted by searching Multiple Em for Motif Elicitation (MEME, Version 5.5.3) website (https://meme-suite.org/meme/tools/meme, accessed on 11 September 2023) with eggplant CATs amino acid sequences. The prediction of the *cis* elements within the promoters of eggplant *CAT* genes was performed by searching PlantCARE (http://bioinformatics.psb.ugent.be/webtools/plantcare/html/, accessed on 11 September 2023) website and then visualized by using TBtools software. The tertiary structure of eggplant CATs was predicted by searching the SWISS-MODEL website (https://swissmodel.expasy.org/interactive, accessed on 11 September 2023) with the build model function.

### 4.5. Collinearity Analysis

The collinearity relationships of eggplant CATs were analyzed and visualized by using TBtools software with the Multiple Collinearity Scan toolkit (MCScanX) functional modules [67]. The duplicated genes of eggplant CTAs with their homologs in tomato and Arabidopsis were performed by using TBtools software with the Advanced Circos functional module.

### 4.6. Plant Materials and Growth Conditions

The seeds of cultivated eggplant ML41 were packaged with clean gauze and incubated in the 55 °C water bath for 15 min, and then were socked in the ddH_2_O at room temperature overnight. The seeds were sowed in the nutrient soil [peat moss: perlite, 2:1 (*v*/*v*)] and placed in the illumination incubator for germination. The eggplant seedlings were transmitted into plastic pots (7 cm × 7 cm) with nutrient soil. The *Nicotiana benthamiana* seeds were sowed on wet filter paper and placed in the illumination incubator for germination. The seedlings of *Nicotiana benthamiana* were transmitted in the plastic pot (7 cm × 7 cm) with nutrient soil. Both the seedlings of eggplant and *Nicotiana benthamiana* were grown in the illumination incubator with 25 °C, 60% relative humidity, 16 h light/8 h dark photoperiod.

### 4.7. Salt Stress, Dehydration, High Temperature, Low Temperature, ABA, and H_2_O_2_ Treatment

The 4-to-6 leaf-stage eggplants were pulled out gently from the soil, and the eggplant roots were washed with clean tap water. The eggplant roots were soaked into the Hoagland nutrient solution for 2 days, and then the roots were respectively soaked in 200 mM NaCl, 100 μM ABA, or 1 mM H_2_O_2_ solution. The treated roots were harvested at the time points of 0, 2, 6, 12, 24, and 48 h post-treatment and frozen in liquid nitrogen. For dehydration stress treatment, the cleaned eggplant roots were dried with filter paper and placed on the bench. The roots were harvested at the time points of 0, 0.5, 1, 3, 6, and 9 h post treatment. For HT or LT stress treatment, the eggplants were respectively placed in the illumination incubator with 43 °C or 4 °C. The treated leaves were harvested in the 2 mL centrifuge tube with 3 small steel balls at the time points of 0, 0.5, 1, 3, 6, and 12 h post-treatment. 

### 4.8. Plant Total RNA Extraction, cDNA Synthesis and RT-qPCR Analysis

The treated roots or leaves were ground into powder by a plant tissue crusher under low-temperature conditions, and the total RNA was extracted by using FastPure Universal Plant Total RNA Isolation Kit (RC411-01, Vazyme, Nanjing city, China). The mRNA was reversely transcribed into cDNA by using HiScript III RT SuperMix for qPCR (+gDNA wiper) (R323-01, Vazyme, China). For real-time quantitative PCR (RT-qPCR) analysis, the ChamQ Universal SYBR qPCR Master Mix (Q711-02, Vazyme, China) and the specific primer pairs listed in Appendix A were used to carry out the RT-qPCR assay to detect the relative transcript expression levels of target genes. *SmActin* (Smechr1100649) was used as the reference gene to normalize the transcript expression levels of target genes. Three biological replications were used, and the relative transcript expression levels of target genes were analyzed by the 2^−ΔΔCT^ method [68].

### 4.9. Vector Construction

For subcellular localization, the full-length open reading frames (ORF) of *SmCAT1*, *SmCAT2*, *SmCAT3*, and *SmCAT4* were respectively amplified by PCR assay with the specific primer pairs and then cloned into the plant overexpression vector pBinGFP2 linearized by DNA restriction endonuclease *Sma* Ⅰ using ClonExpress II One Step Cloning Kit (C112-01, Vazyme, China). For prokaryotic expression, the full-length ORF of *SmCAT4* was amplified by PCR with the specific primer pairs and then cloned into the multiple clone site *Sma* Ⅰ in the pGEX-6P-1 vector by using ClonExpress II One Step Cloning Kit (C112-01, Vazyme, Nanjing city, China). For the VIGS assay, the 300 bp specific DNA fragment of *SmCAT4* was amplified by PCR and then cloned into entry pDONR207 vector by BP reaction, and then transmitted into the destination vector pTRV2 by LR reaction. The primer pairs used for vector construction in this study were listed in Appendix A.

### 4.10. Agrobacterium tumefaciens Cultivation and Infiltration and Subcellular Localization Analysis

*Agrobacterium tumefaciens* strain GV3101 containing 35S:*SmCAT1-GFP*, 35S:*SmCAT2-GFP*, 35S:*SmCAT3-GFP*, or 35S:*SmCAT4-GFP* constructs were cultivated in the liquid Luria-Bertani (LB) medium with 50 μg/mL kanamycin and 50 μg/mL rifampicin antibiotics at the conditions of 28 °C, 200 rpm, overnight. The bacteria solution was centrifuged at 6000 rpm for 5 min at room temperature, and then the bacterial cells were resuspended by infiltration buffer (10 mM MES, 10 mM MgCl_2_, 200 mM acetosyringone, pH = 5.4) to adjust OD_600_ to 0.8. The bacterial solution was infiltrated into the leaves of *Nicotiana benthamiana* using a disposable sterilized syringe without a needle. The infiltrated plants were cultivated in the illumination incubator. After 48 h, the signaling in the epidemic cells of *Nicotiana benthamiana* leaves was observed by laser scanning confocal microscope (TCS SP8, Leica Microsystems, Weztlar, Germany).

### 4.11. VIGS Assay

The *Agrobacterium tumefaciens* GV3101 cells harboring pTRV1, pTRV2:*00*, pTRV2:*SmPDS*, or pTRV2:*SmCAT4* constructs were cultivated in liquid LB medium overnight and then adjusted OD_600_ into 0.8 by infiltration buffer. The bacterial cells containing the pTRV1 vector were mixed with the GV3101 cells carrying pTRV2:*00*, pTRV2:*SmPDS*, or pTRV2:*SmCAT4* constructs at 1:1 ratio, and the mixtures were tenderly incubated in the 28 °C shaker at 60 rpm for 3 h. The Agrobacterium solution was infiltrated into the cotyledon of 2-to-3 leaf-stage eggplants. The infiltrated eggplant seedlings were placed in the illumination incubator without light at 20 °C for 48 h. The treated seedlings grew in the illumination incubator for 3 weeks until the leaves of TRV:*SmPDS* plants turned white.

### 4.12. Prokaryotic Expression and Purification of Recombinant Protein

The *Escherichia coli* strain BL21 (DE3) harboring pGEX-6P-1:*SmCAT4* (containing a GST protein tag) constructs or empty vector pGEX-6P-1 were cultivated in the liquid LB medium with 100 μg/mL ampicillin at 37 °C, 200 rpm until the OD_600_ of the bacterial solution to 0.6, and then added 2 mM isopropyl-β-d-thiogalactoside into the bacterial solution. The bacterium was cultivated at 18 °C, 200 rpm overnight. Sodium dodecyl sulfate polyacrylamide gel electrophoresis (SDS-PAGE) and Coomassie bright blue staining assays were performed to detect the target proteins, whether they were expressed or not. For the purification of recombinant proteins, the bacterium cells were broken by an ultrasonic cell crusher. The supernatant was incubated with the magnetic beads of BeaverBeads™ GSH kit (70601-5, Suzhou Beaver Biomedical Engineering Co., LTD, Suzhou city, China) at 4 °C for 1 h, and the target proteins on the magnetic beads were eluted by elution buffer after washing.

### 4.13. Recombinant CAT Enzyme Assay

The detection of the purified recombinant SmCAT4-GST or GST proteins was performed following a previously described study [69].

### 4.14. Physiological Index Measurement

The eggplant roots of TRV:*00* or TRV:*SmCAT4* plants treated with salt stress were harvested, and the physiological index measurement of CAT enzyme activity and H_2_O_2_ content were performed following previously described studies [69,70].

## 5. Conclusions

In this study, we identified four *CAT* genes in the eggplant genome and further analyzed their sequences, structures, and expressions in eggplant. Phylogenetic classification divided CAT proteins into subgroups Ⅰ and Ⅱ. Further sequences and structure analysis of *CAT* genes revealed a high degree of conservation among the *CAT* gene family members in eggplant. Based on the result of different gene expressions of *CAT* genes under abiotic stress such as salt, drought, cold, and high temperature stress, we found that *CAT* genes may play an important role in eggplant response to these abiotic stresses. Moreover, eggplant SmCAT1-3 proteins were localized in the peroxisome, and SmCAT4 was localized in the cytomembrane and nucleus. *SmCAT4* expression was induced by salt, dehydration, LT, and HT treatment. VIGS assay revealed that silencing of *SmCAT4* could compromise the tolerance level of eggplant to salt stress. Our data provide new insight into a comprehensive understanding of the *CAT* gene family and help further explore the functions of *CAT* family members in eggplant response to various abiotic stresses.

## Figures and Tables

**Figure 1 ijms-24-16979-f001:**
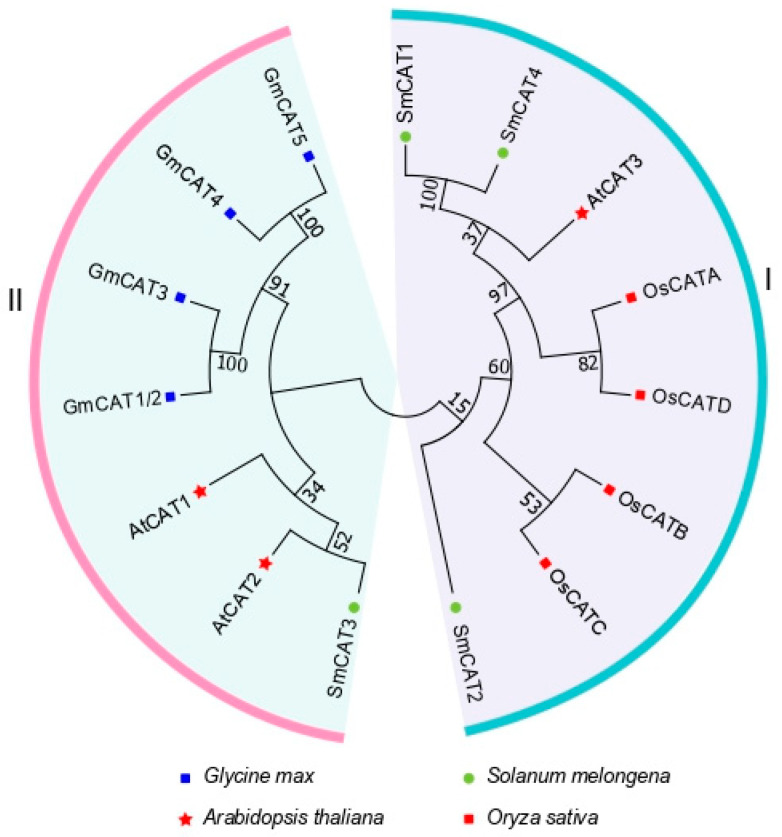
Analysis of phylogenetic relationship of *CAT* gene family members. Phylogenetic tree was generated to analyze the phylogenetic relationship between eggplant *CAT* gene family members and the *CAT* members from *Arabidopsis thaliana*, rice (*Oryza sativa*), and soybean (*Glycine max*) by MEGA 7.0 software.

**Figure 2 ijms-24-16979-f002:**
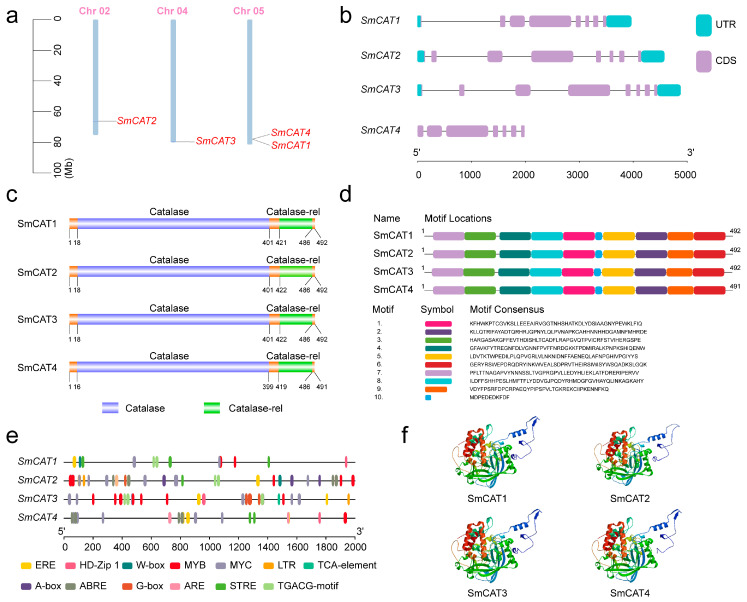
Analysis of chromosomal assignment, gene structures, conserved domain and motifs, *cis*-elements, and tertiary structure of *catalase* gene family members in eggplant. (**a**) Distribution of eggplant *CAT* genes on chromosomes. (**b**) The gene structure of eggplant *CAT* genes predicted by TBtools software. UTR, untranslated region; CDS, coding sequence. (**c**) The conserved domains of eggplant CAT proteins predicted by SMART website. (**d**) Analysis of conserved motifs of eggplant CAT proteins predicted by MEME website. (**e**) Analysis of *cis* elements within the promoters of eggplant *CTA* genes by searching PlantCARE website. (**f**) The tertiary structure of eggplant CAT proteins including SmCAT1, SmCAT2, SmCAT3, and SmCAT4 by searching the SWISS-MODEL website.

**Figure 3 ijms-24-16979-f003:**
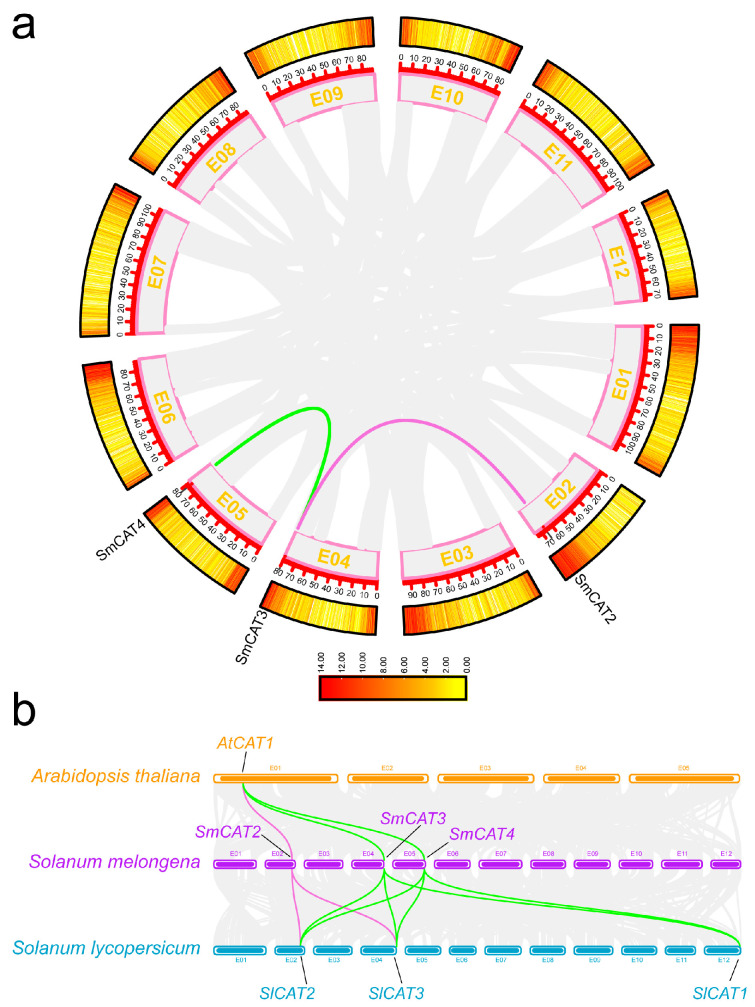
The collinearity analysis among the eggplant *CAT* genes (**a**) and of eggplant CAT genes with *Arabidopsis thaliana CAT* genes and tomato *CAT* genes (**b**). In (**a**), the gray lines represent all co-linear gene pairs in eggplant, and the green and pinkish red lines respectively indicate co-linear gene pairs *SmCAT3* and *SmCAT4* as well as *SmCAT2* and *SmCAT3*. In (**b**), the gray lines represent all co-linear gene pairs in eggplant, *Arabidopsis thaliana*, and tomato, and the pinkish red, green, and red lines respectively represent the collinear gene pairs of *SmCAT2*, *SmCAT3,* and *SmCAT4* in eggplant, *Arabidopsis thaliana,* and tomato, respectively.

**Figure 4 ijms-24-16979-f004:**
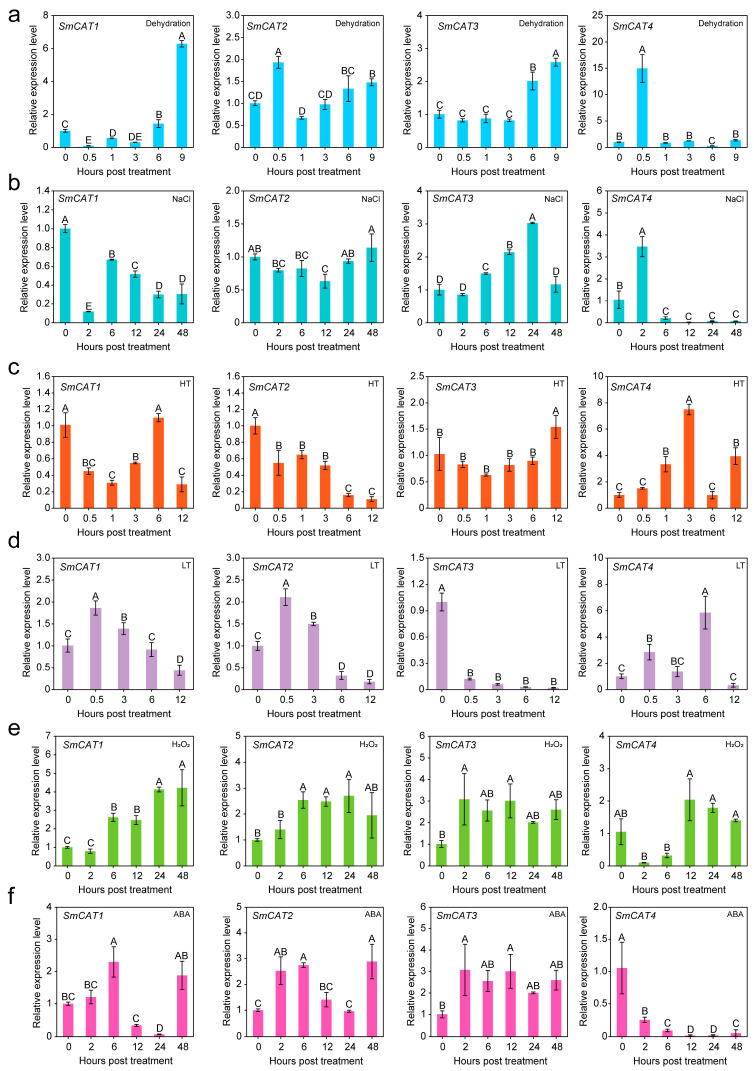
Analysis of the relative transcript expression levels of *SmCAT1*, *SmCAT2*, *SmCAT3*, and *SmCAT4* under the condition of dehydration (**a**), 200 mM NaCl (**b**), high temperature (HT, 43 °C) (**c**), low temperature (LT, 4 °C) (**d**), H_2_O_2_ (**e**), and 100 μM abscisic acid (ABA) (**f**) treatment. Data indicate the means ± SD from three biological repeats. Different upper letters indicate significant differences, as determined by Fisher’s protected LSD test (*p* < 0.01).

**Figure 5 ijms-24-16979-f005:**
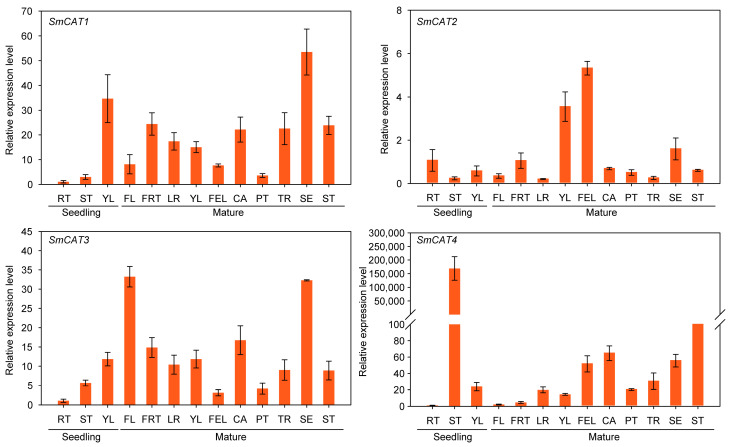
Analysis of the relative transcript expression levels of four eggplant *CAT* genes in various organs of eggplant seedlings and mature plants. RT, root; ST, stem; YL, young leaf; FL, flower; FRT, fruit; LR, lateral root; FEL, fully expanded leaf; CA, carpopodium; PT, petiole; TR, tap root; SE, sepal. Data indicate the means ± SD from three biological repeats.

**Figure 6 ijms-24-16979-f006:**
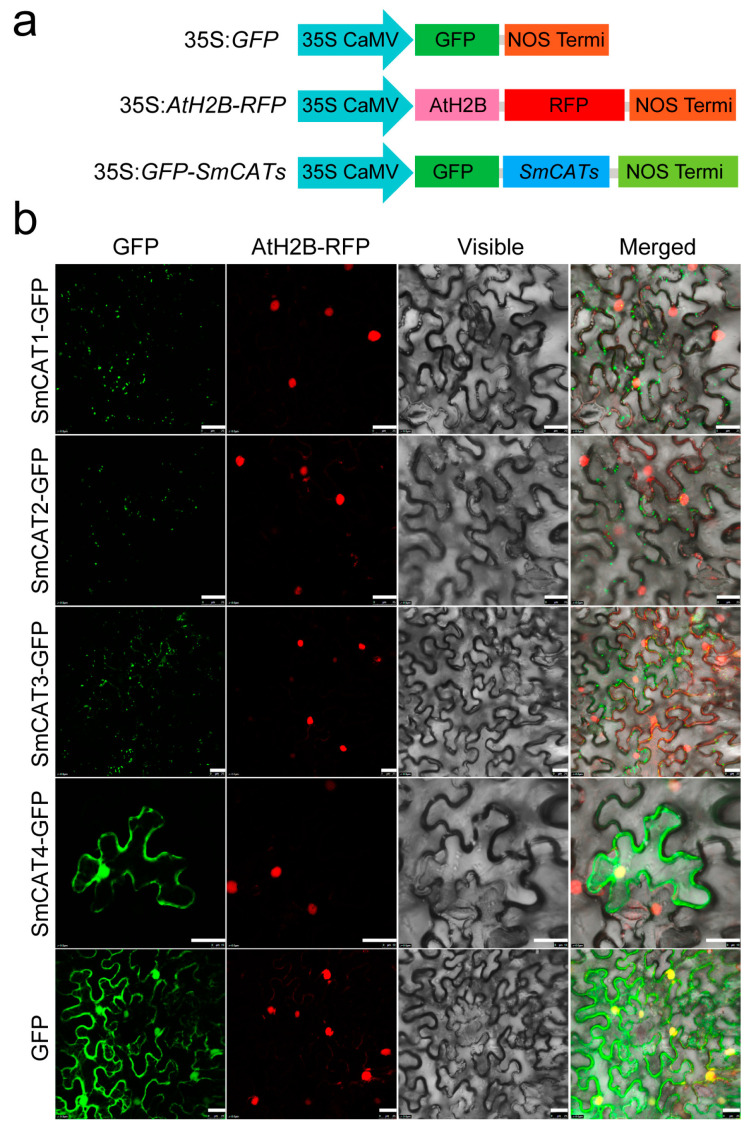
Analysis of subcellular localization of eggplant CAT proteins. (**a**) The schematic diagrams of 35S:*GFP* (empty vector), 35S:*AtH2B-RFP* (a subcellular localization marker of nucleus), and 35S:*GFP-SmCATs* structures. (**b**) Subcellular localization of eggplant CAT proteins in the epidermic cells of *Nicotiana benthamiana* leaves. Bar = 25 μm.

**Figure 7 ijms-24-16979-f007:**
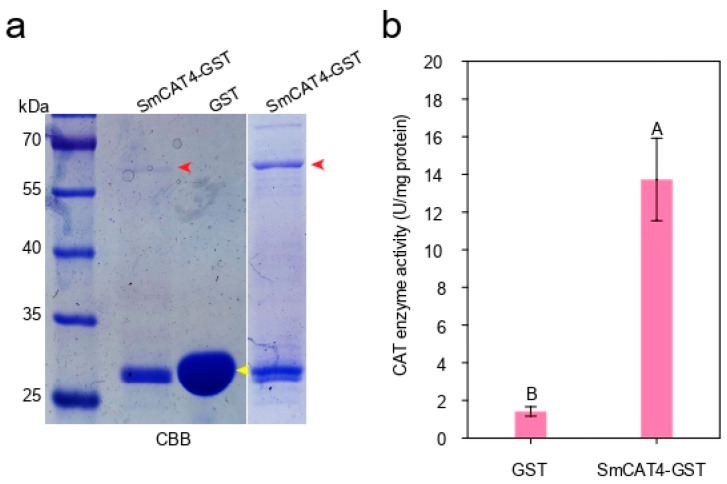
Analysis of CAT enzyme activity of SmCAT4-GST in vitro. (**a**) Analysis of purification of recombinant protein SmCAT4-GST and GST. The recombinant protein SmCAT4-GST and GST were separated by SDS-PAGE assay and the gel was then stained by Coomassie brilliant blue (CBB) solution and de-stained by a destainer. Red arrows indicate SmCAT4-GST protein, and yellow arrow represent GST protein. (**b**) Detection of CAT enzyme activity of SmCAT4-GST and GST protein in vitro. Data indicate the means ± SD from three biological repeats. Different upper letters indicate significant differences, as determined by Student’s *t*-test (*p* < 0.01).

**Figure 8 ijms-24-16979-f008:**
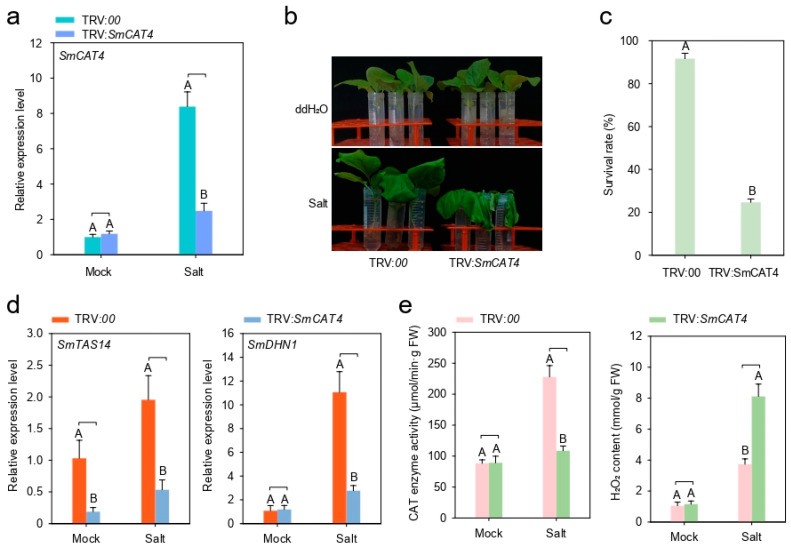
Silencing of *SmCAT4* enhanced susceptibility of eggplant against salt stress. (**a**) Analysis of silencing efficiency of *SmCAT4* by performing RT-qPCR assay. (**b**) Enhanced salt sensitivity of *SmCAT4*-silenced eggplants. The roots of the control and *SmCAT4*-silenced plants were soaked into 200 mM NaCl solution or ddH_2_O for 2 days. (**c**) Calculation of survival rate of the control and *SmCAT4*-silenced plants treated with 200 mM NaCl solution at 48 h post treatment. (**d**) Analysis of transcript expression levels of salt stress defense related marker genes *SmTAS14* and *SmDHN1* in the roots of *SmCAT4*-silenced or control plants at 24 h post salt stress treatment. (**e**) Measurement of CAT enzyme activity and H_2_O_2_ content in the roots of *SmCAT4*-silenced or control plants at 24 h post salt stress treatment. In (**a**,**c**–**e**), the data indicate the means ± SD from three biological repeats. Different upper letters indicate significant differences, as determined by Student’s *t*-test (*p* < 0.01).

**Table 1 ijms-24-16979-t001:** Physicochemical properties of CAT proteins of *Solanum melongena*.

Gene Name	Base Number	Amino Acid Number	Molecular Weight	Instability Index	Aliphatic Index	Grand Average of Hydropathicity	Average of Hydropathicity	Theoretical pI	Subcellular Localization
*SmCAT1*	1479	492	56,594.07	39.78	71.91	−0.515	−0.515	6.86	Peroxisome
*SmCAT2*	1479	492	56,996.23	40.25	68.94	−0.582	−0.582	6.88	Peroxisome
*SmCAT3*	1479	492	56,934.43	38.18	70.35	−0.552	−0.552	6.80	Peroxisome
*SmCAT4*	1476	491	56,545.00	38.37	71.87	−0.575	−0.575	7.31	Peroxisome

## Data Availability

Data will be made available on request.

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
