# Peer review of "Genome-Wide Identification of Catalase Gene Family and the Function of SmCAT4 in Eggplant Response to Salt Stress"

_ijms, 2023, doi:10.3390/ijms242316979_

Round 1

Reviewer 1 Report

Comments and Suggestions for Authors

The study is amide to investigate genome-wide identification of catalase gene family and the function of SmCAT4 in eggplant response to salt stress. The study design is acceptabel. The study contains some valuable results that can be considered for possible publication after suitable revisions.

Suggestions:

Table 1 should be placed at first mention (arond page 2).

Figure 1: Give in full CAT in the title.

Figure 2: Give in full in the title: CAT. Please increase the size of 'f' figure.

Figure 4: Give in full in the title: CAT, ABA.

Table 1. Pyhsiochemical ... CAT-give in full in the title.

Figure 8e. H2O2 - please correct.

References need to be fixed in some points.

L538-549: Missing journal issues and page numbers.

L651: H(2)O(2) ???

Author Response

Table 1 should be placed at first mention (arond page 2).

Response: thank you very much. We placed Table 1 at the first mention, please see in line 94 in the revised manuscript.

Figure 1: Give in full CAT in the title.

Response: thank you very much. We added the full CAT in the Figure 1 legend title, please see in line 767 in the revised manuscript.

Figure 2: Give in full in the title: CAT. Please increase the size of 'f' figure.

Response: thank you very much. We modified the Figure 2 legend title, please see in 772-773 lines in the revised manuscript. In addition, we increased the size of ‘f’ figure, please see the revised Figure 2f.

Figure 4: Give in full in the title: CAT, ABA.

Response: thank you very much. We modified the Figure 4 legend title, please see in 792-793, 795 lines in the revised manuscript.

Table 1. Pyhsiochemical ... CAT-give in full in the title.

Response: thank you very much. We modified the Table 1 title, please see in line 833 in the revised manuscript.

Figure 8e. H2O2 - please correct.

Response: thank you very much. We corrected the H2O2 to H2O2 in the revised Figure 8e

References need to be fixed in some points.

Response: thank you very much. We fixed the references, please see the revised references in the revised manuscript.

L538-549: Missing journal issues and page numbers.

Response: thank you very much. We added the journal issues and page numbers, please see the modified references in the revised manuscript.

L651: H(2)O(2) ???

Response: thank you very much. We corrected the H(2)O(2) to H2O2 , please see it in line 714 in the revised manuscript.

Reviewer 2 Report

Comments and Suggestions for Authors

This study by Shen et al. identifies and characterises four catalase (CAT) genes in the eggplant genome, highlighting their conservation and differential expression under abiotic stresses like salt, drought, cold, and heat. Specifically, the research emphasises the unique role of SmCAT4, which, unlike other CAT proteins localised in peroxisomes, is found in the cytomembrane and nucleus. SmCAT4 is up-regulated by various stress conditions and is crucial for eggplant's salt stress response, underscoring its potential in enhancing stress tolerance in eggplants. Overall, the study has been well performed and, for the most part, well-written. However, I have a few minor concerns that the authors should address:

Line 200: “We further investigated the transcript expression levels of eggplant CATs in the different tissues of seedlings and mature (Figure 5)” is incomplete.

Entire manuscript needs language editing by a native speaker.

Few examples from the Discussion section:

Line 279: The phrase "serious damaged of the stability" should be "serious damage to the stability." The current form is grammatically incorrect.

Line 354: The phrase "Although we predicted the subcellular localization of these four CAT proteins, which localized in the peroxisome, we provided evidence that eggplant CAT1-3 proteins located in the peroxisome, while SmCAT4 localized in the cytomembrane and nucleus" is awkwardly structured.

Line 287: "in plant involving in the regulatory of salt stress resistance…" Should be “…involved in the regulation of salt stress …”

Line 298: The word "excess" should be either “in excess” or "excessive" for grammatical correctness. The sentence should read, "When ROS production is excessive [or, in excess], uncontrolled oxidation ultimately leads to cellular damage and even cell death."

Comments on the Quality of English Language

Entire manuscript needs language editing by a native speaker.

Few examples from the Discussion section:

Line 279: The phrase "serious damaged of the stability" should be "serious damage to the stability." The current form is grammatically incorrect.

Line 354: The phrase "Although we predicted the subcellular localization of these four CAT proteins, which localized in the peroxisome, we provided evidence that eggplant CAT1-3 proteins located in the peroxisome, while SmCAT4 localized in the cytomembrane and nucleus" is awkwardly structured.

Line 287: "in plant involving in the regulatory of salt stress resistance…" Should be “…involved in the regulation of salt stress …”

Line 298: The word "excess" should be either “in excess” or "excessive" for grammatical correctness. The sentence should read, "When ROS production is excessive [or, in excess], uncontrolled oxidation ultimately leads to cellular damage and even cell death."

Author Response

Reviewer2

This study by Shen et al. identifies and characterises four catalase (CAT) genes in the eggplant genome, highlighting their conservation and differential expression under abiotic stresses like salt, drought, cold, and heat. Specifically, the research emphasises the unique role of SmCAT4, which, unlike other CAT proteins localised in peroxisomes, is found in the cytomembrane and nucleus. SmCAT4 is up-regulated by various stress conditions and is crucial for eggplant's salt stress response, underscoring its potential in enhancing stress tolerance in eggplants. Overall, the study has been well performed and, for the most part, well-written. However, I have a few minor concerns that the authors should address:

Line 200: “We further investigated the transcript expression levels of eggplant CATs in the different tissues of seedlings and mature (Figure 5)” is incomplete.

Response: thank you very much. We corrected this mistake, please see it in line 185 in the revised manuscript.

Entire manuscript needs language editing by a native speaker.

Response: thank you very much. Our manuscript was edited by a native speaker. Please see our revised manuscript.

Few examples from the Discussion section:

Line 279: The phrase "serious damaged of the stability" should be "serious damage to the stability." The current form is grammatically incorrect.

Response: thank you very much. We corrected this mistake, please see it in line 227 in the revised manuscript.

Line 354: The phrase "Although we predicted the subcellular localization of these four CAT proteins, which localized in the peroxisome, we provided evidence that eggplant CAT1-3 proteins located in the peroxisome, while SmCAT4 localized in the cytomembrane and nucleus" is awkwardly structured.

Response: thank you very much. We corrected this mistake, please see it in line 308 in the revised manuscript.

Line 287: "in plant involving in the regulatory of salt stress resistance…" Should be “…involved in the regulation of salt stress …”

Response: thank you very much. We corrected this mistake, please see it in line 236 in the revised manuscript.

Line 298: The word "excess" should be either “in excess” or "excessive" for grammatical correctness. The sentence should read, "When ROS production is excessive [or, in excess], uncontrolled oxidation ultimately leads to cellular damage and even cell death."

Response: thank you very much. We corrected this mistake, please see it in lines 38, 246 in the revised manuscript.